

# A scoping review of tropical pioneer trees' roles for restoration and conservation management: *Harungana madagascariensis* (Hypericaceae) a widespread African species as a model

François M.M.P. Baguette[1], Cláudia Baider[2] and F.B. Vincent Florens[1]

[1] Tropical Island Biodiversity, Ecology and Conservation Pole of Research, Department of Biosciences and Ocean Studies, Faculty of Science, University of Mauritius, Le Réduit, Mauritius
[2] The Mauritius Herbarium, Agricultural Services, Ministry of Agro-Industry, Food Security, Blue Economy and Fisheries, Le Réduit, Mauritius

Corresponding author
François M.M.P. Baguette,
francois.baguette@umail.uom.ac.mu

## ABSTRACT

**Background**. Globally, biodiversity is declining rapidly, including tropical forests ecosystems in particular. To stop and reverse this trend, governments worldwide signed up to international agreements and initiatives, but success to date has been limited. In this context, reviewing pioneer trees' ecology, particularly the most widespread species, can help gauge their pros and cons and guide their judicious use for cost-effective ecological restoration projects.

**Objectives**. This study aims to review the potential of pioneer tree species for biodiversity conservation and forest restoration and identify eventual knowledge gaps, using a widespread species from Africa, *Harungana madagascariensis* Lam. (Hypericaceae), as a model. Our specific objective was to synthetize information on the distribution and habitat of *H. madagascariensis*; its documented interspecific ecological interactions; and its potential for forest restoration.

**Methodology**. A scoping review was conducted using multiple databases to identify relevant papers, supplemented by the Global Biodiversity Information Facility database (GBIF) to extract distribution records of *H. madagascariensis*. Following the PRISMA screening process for scoping reviews, 398 publications and 4,379 records from GBIF (2023) were used in the final analysis out of a total of 1,159 unique articles and 5,230 records originally retrieved.

**Results**. We show that *H. madagascariensis*, which is native to tropical Africa, Madagascar, and some islands of the Mascarenes, is a key component of young secondary forests, wetland areas, and grasslands. At least 125 species were found to interact directly with *H. madagascariensis*, including through commensalism, mutualism, and herbivory. It is recognized as a tool for restoration regionally, and considered as invasive in Australia where it has been introduced and, by some, in Mauritius where it is native. The benefits it provides for restoration include its capacity to improve degraded soil fertility, its ability to compete with invasive alien species mostly due to its heliophilous and fast-growing nature, and its good nurse tree potential along with its ecological interactions that support numerous species including threatened ones.

**Conclusion**. The widespread African pioneer tree *H. madagascariensis* plays a critical role in vegetation dynamic and holds great potential for fostering forest restoration and

biodiversity conservation in its range of nearly 13 M km$^2$. Its greater use in restoration projects could significantly accelerate ecological restoration, decrease its costs, and increase benefits to biodiversity, leading to larger areas being restored, contributing effectively to national and international objectives. However, a number of aspects deserve further studies, such as the species' role in multitrophic interactions and its precise interactions, and their strengths, with species in each of its specific geographical contexts and through different temporal scales.

## INTRODUCTION

Globally, biodiversity is decreasing rapidly driven by a variety of human activities and impacts (*Baillie, Hilton-Taylor & Stuart, 2004*; *Vie, Hilton-Taylor & Stuart, 2009*; *Barnosky et al., 2011*). Without further efforts to counteract those drivers, it has been shown that this trend will continue, reaching peak losses of 12% for global species extinction and 63% for wildlife population density by 2100 (*Leclère et al., 2020*). In the past 20 years, remarkable progress has been made towards understanding how the loss of biodiversity affects the functioning of ecosystems and thus affects society and human well-being (*Cardinale et al., 2012*; *Pimm et al., 2014*). Changes to land cover only from the past twenty years have reduced the value of the annual flow of ecosystem services by USD 4–20 trillion/yr (*Costanza et al., 2014*) and in this context, it is important to promote greater efforts of conservation and ecological restoration.

From all ecosystems, tropical forests contain some of the greatest concentrations of biodiversity on the planet. But over the past two decades, up to 80 million hectares of tropical forest have been destroyed, and a further 3.7 million hectares has been lost in 2023 (*Weisse, Goldeman & Carter, 2024*). As a result, secondary forests are on the rise and estimated to comprise >60% of tropical forest landscapes (*Chazdon, 2014*), gradually replacing primary forests throughout the tropics (*Wright, 2005*). Oceanic islands in particular are most subjected to this situation (*Rull, 2020*) due to their physical isolation that prevents recolonization from other potential forest sources. This is epitomized by Mauritius, a 1,865 km$^2$ volcanic island in the Western Indian Ocean, which lost over 95% of its original vegetation cover (*Hammond et al., 2015*), and where despite the implementation of internationally recognized conservation efforts and the creation of protected areas, native forests and biodiversity continue to decline (*Florens & Baider, 2013*; *Florens et al., 2017*; *Bissessur et al., 2023*).

To stop and reverse the loss of biodiversity, including tropical forest cover, governments worldwide have subscribed to multiple international agreements, including the Convention on Biological Diversity (CBD) and the United Nations Sustainable Development Goals (SDG) frameworks (*Convention on Biological Diversity, 2011*; *United Nations, 2015*). In 2022, the UN developed a new Global Biodiversity Framework (https://www.cbd.int/gbf)

setting new targets for effective restoration of degraded areas (*e.g.*, Target 2), including terrestrial ecosystems. To achieve these, international initiatives like the 2011 Bonn Challenge and the 2021–2030 UN Decade on Ecosystem Restoration were launched but success to date has been limited (*Hoffmann et al., 2010*; *Pimm et al., 2014*; *Tittensor et al., 2014*; *Waring, 2024*). As those ambitions could even be impaired by projects that ignore key principles such as inclusion of communities, equitable benefit sharing, and the use of evidence-based management (*Florens & Baider, 2013*; *Höhl et al., 2020*), there is still much room to improve and upscale restoration projects worldwide (*Suding, 2011*).

One of the tools that has been recommended to facilitate forest restoration initiatives is the adequate selection of species and the use of pioneer trees (*Viani et al., 2015*; *Spînu et al., 2023*). Pioneer species have been recognized as a potential tool to foster ecological restoration because of their rapid growth and maturation, capacity to rapidly form a canopy, ability to grow on relatively poor soils, and their large production of seeds. In addition, some may serve as useful nurse trees (*Binggeli, 1989*; *Jones, 2008*) or even suppressors of heliophilous invasive alien plants where these can be problematic (*Otuoma et al., 2020*). However, those benefits have rarely been quantified and as a result, pioneer trees are still often undervalued or disregarded by conservation managers who sometimes prefer to plant species without much considerations of habitat requirements, especially on islands (*Baguette et al., 2022*) or even control them to open space for planting other species (*Florens & Baider, 2013*) or because they are perceived as invasive (*Strahm, 1993*; *Ragen, 2007*; *Swinfield et al., 2016*) despite the absence of evidence for that and their potential of being an asset to lower forest restoration costs and upscale restoration projects.

Based on this context, a scoping review was conducted to quantify the potential of pioneer tree species for biodiversity conservation and forest restoration, using a widespread species from Africa, *Harungana madagascariensis* Lam. (Hypericaceae), as a model, as well as identify any existing gaps in knowledge. Our objectives were to synthetize the literature on (1) the distribution and habitat of *H. madagascariensis*; (2) its documented interspecific ecological interactions; and (3) its potential for forest restoration, specifically to explore the extent to which pioneer tree species may support ecological restoration and biodiversity conservation. We compiled information from all its range, including where it has been introduced, and summarized the current state of knowledge on *H. madagascariensis*. The study thus allowed us to examine the influence that a pioneer tree species may have on biodiversity, and define its role in forest restoration. This information can help to support forest restoration strategies, especially in tropical regions and the African continent, which is relatively less studied compared to other regions of the world, and help identify gaps in the literature which may guide further research.

## METHODOLOGY

### Literature selection

The review protocol followed the PRISMA for Scoping Reviews (PRISMA-ScR) guidelines (*Tricco et al., 2018* see Fig. 1 for PRISMA flow diagram and Table S1 for PRISMA checklist). The search strategy was drafted by the first author and further refined
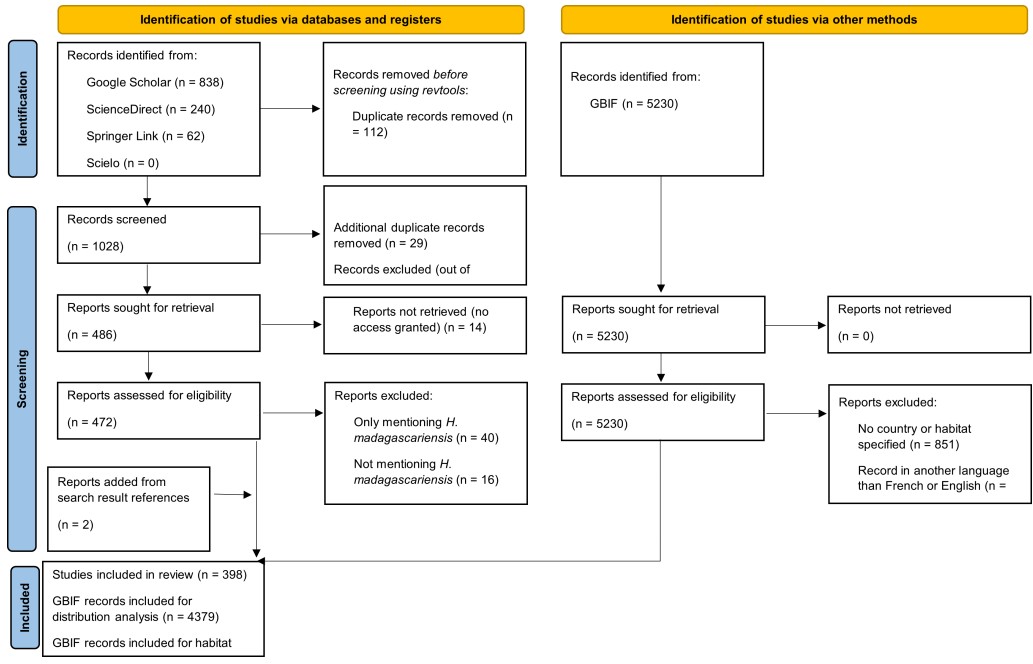

**Figure 1** PRISMA diagram for the progression of papers included in this scoping review.

through team discussion. To identify potentially relevant documents on the ecology of *H. madagascariensis*, its distribution, role for biodiversity and restoration potential, four scientific databases, namely Google Scholar, ScienceDirect, SpringerLink, and Scielo were searched in November 2023 by the first author of this study (FB). The search terms ''*Harungana madagascariensis*'' AND ''ecology'' OR ''habitat'' OR ''host'' OR ''herbivory'' OR ''dispersal'' OR ''forest succession'' OR ''pollination'' OR ''phylogeny'' OR ''restoration'' were used in all databases and the same string of search terms was used with all synonyms of the species and alternative spellings. The final search results were exported into Zotero, and duplicates were removed by FB. The electronic database search was supplemented by searching for records of *H. madagascariensis* in the Global Biodiversity Information Facility database (GBIF) and relevant articles found from the references of the search results have also been included, as well as 17 additional articles retrieved from preliminary searches. This resulted in a total of 1,159 unique articles and 5,230 records (*GBIF, 2023*).

Our intention was to review all the literature available and therefore no restriction on the type of publication was applied. To be included in the analysis, papers and GBIF records had to meet the following inclusion criteria: (1) specifically address at least one of the above-mentioned objectives; and (2) be written in English or French. Papers focusing on other subjects such as medicine, ethnobotany, or pharmacology were excluded, as well as papers only referring to other references or only mentioning *H. madagascariensis* as example. We used a two-phase screening process to remove irrelevant material, first based
on title and abstract and then on full-text screening. Prior to screening, 112 duplicates were removed using the R package "revtools" (*Westgate, 2019*).

For the first phase of the screening, two coders (the first author, FB, and second author, CB) both assessed the first 579 abstracts of the search results (55.1%). Based on these 579 abstracts, a high level of inter-rater reliability was found between coders for accept/reject decisions ($\chi^2 = 0.016$, $p = 0.9005$), hence the remaining abstracts were coded by FB. To ensure the reliability of the criteria for full-text screening, FB and CB both assessed 86 full-texts out of the 283 available from the initial inter-screening phase (30.4%). Disagreement was found for only three articles (3.5%), of which two were due to coding mistake by one of the coders. Hence, as there was a near complete inter-rater reliability ($\chi^2 = 0.068$, $p = 0.7946$) between coders for accept/reject decisions on the basis of these 86 full-texts, the remaining full-texts were coded by FB. Any disagreements over the relevancy of a given article in either phase were resolved with the full screening team through consensus among the authors. After review, 398 papers were included in the final analysis (Table S2), as well as 4,379 records from GBIF (Table S3).

## Data collection

A data-charting form was jointly developed *a priori* to determine which variables to extract. All data were then extracted by the first author, verified by the others, and validated together before analysis. Any disagreements over the relevance of the data extracted were resolved through consensus among the authors. First, general data on article type (*e.g.*, peer-reviewed article, review, book, book chapter, conference paper, Master thesis, PhD thesis, project report, colloquium article, and workshop proceedings) have been recorded for each search result as well as the geographic scope it was covering. For scientific articles, studies, and project reports, the locations in which the works were conducted were recorded. For reviews and more general documents, the geographic range covered was recorded only when clearly mentioning the presence of *H. madagascariensis*. The number of publications per country was then compiled, as well as the percentage of each publication type per country. This information was used as proxy to evaluate the research effort devoted to *H. madagascariensis* across its range, and identify areas where knowledge gaps exist. All data were consolidated into Microsoft Excel and analyses were done using the same software. Maps were produced in QGIS 3.28.15 Firenze.

We then made a detailed assessment of the geographic range of *H. madagascariensis*, extracting the locations where it was recorded from each paper, as well as the habitat characteristics of where it was observed such as ecosystem type, elevation range, annual rainfall, soil type, and temperature range whenever clearly defined. Multi-location papers that dealt with more than one study area were only included where it was possible to assign *H. madagascariensis* to specific locations. Additionally, records from GBIF (*GBIF, 2023*) that clearly mentioned either the location or the habitat or any of the above-mentioned habitat characteristics from which *H. madagascariensis* had been observed, were included in the analysis. Finally, information on its position within forest succession as well as its status (native or introduced) in each country, and whether the species was considered invasive or not, was also recorded from all search results and *GBIF (2023)* records.

![PeerJ]

**Table 1  Definitions and study counts of all types of ecological interactions involving *Harungana madagascariensis* and included in this review.** Flower visitation has been defined as potential mutualism as it could be either mutualistic (leading to pollination) or not (if nectar is consumed without leading to pollination). Frugivory has been defined as predation or mutualism as it could either lead to seed dispersal (mutualism) or seed damage (predation). The list of all publications linked to each interaction is available in Table S2.

| Interaction type | | Definition | Number of studies |
|---|---|---|---|
| Ammensalism | | Interaction in which one organism is unaffected while the other is suffering (*e.g.*, allelopathy) | 0 |
| Commensalism | | Interaction in which one organism serves as a host for another one, benefiting one while the other is unaffected (*e.g.*, host for epiphyte and saprophytic fungi) | 6 |
| Competition | | Interaction in which two organisms compete with each other and affect each other negatively | 4 |
| Mutualism | Mycorrhizal association | Symbiotic association between plant roots and fungi | 8 |
| | Myrmecochory | Dispersal of fruits and seeds by ants | |
| Mutualism (potential) | Flower visitation (potential pollination) | Interaction in which one organism visits the flowers of a plant. | 12 |
| Predation | Florivory | Flower consumption, including removal of flower buds, flowers, and inflorescences (complete or incomplete) | 13 |
| | Folivory | Leaf consumption | |
| | Parasitism | Interaction between two living species in which one organism is benefitted at the expense of the other. In this study, interactions with plant, animal or fungal parasites have been included in this category | |
| | Stem damage | Damage to the stem, including puncture damage and meristem removal | |
| Predation or mutualism | Frugivory | Fruit consumption | 29 |
| Undefined | Undefined | Interaction recorded in which one organism is interacting with the other without specifying the nature of interaction | 21 |

Following this, the interspecific ecological interactions in which *H. madagascariensis* was involved were compiled from all results and classified following a general textbook of ecology (*Begon, Townsend & Harper, 2012*). The definitions and study count of all types of interactions included in this review are summarized in Table 1, and the reference of all associated publications made available in Table S2. Study counts apply to all versions for each broad interaction category (for example, florivory, folivory, and stem damage all fall under "herbivory" which is itself a form of "predation"; frugivory on the other hand can be considered either as a form of predation if seeds are destroyed or of mutualism if seed dispersal is promoted and therefore has been listed separately). In addition, the number and list of species linked to each interaction has been recorded, as well as the part of the plant involved in the interaction, whenever this information was made available. In cases where species were mentioned interacting with *H. madagascariensis* without specifying the nature of the interaction, these were recorded as "undefined".

Finally, search results including information on the benefits of *H. madagascariensis* for forest restoration have been compiled and data thereon extracted. Here, forest restoration is understood as any action or project that aims to improve the biodiversity, ecological integrity and provision of services in forest ecosystems. As such, actions such as rewilding, reforestation, afforestation, remediation, rehabilitation, prestoration (restoration that

specifically includes climate change adaptation), or any shift in direction towards a closer-to-nature forest management, was included within the term restoration. Similarly, information on any threat arising from the presence of *H. madagascariensis* for biodiversity and forest restoration was also recorded. All perceived or proven benefits and threats were paired with the specified locations.

## RESULTS

### General search results

In all, 398 search results met all criteria to be included in the final analysis, as well as 4,382 records from GBIF (2023). The majority of the search results included journal articles (244), followed by PhD theses (38), Master theses (37), reports (37), books and book chapters (35), conference papers (four), colloquium article (one), scientific note (one), and workshop proceeding (one). These results ranged from 1935 to 2023 spanning 32 different countries and two (of 14) different biomes. The majority of the results originated from tropical mainland Africa (309), followed by Madagascar (75), and the Mascarenes (nine) where it is native (Fig. 2), and Australia where it has been introduced (five). From *GBIF (2023)*, out of the 4,379 records that were included in the final distribution analysis, only 824 included clear information on the habitat of *H. madagascariensis*. Most GBIF records of the species originated from the Democratic Republic of Congo (782), Gabon (657) and Madagascar (625), with Mauritius (14), Mali (six), Reunion (three), Guinea Bissau (three), and Gambia (one) being the least documented.

From all search results included in the analysis, 58.0% ($N = 231$) contained clear information on habitat characteristics and ecology of *H. madagascariensis* and were used to review its distribution and habitat with the additional 4,382 records from GBIF (2023). Beside these, 23.4% of the results ($N = 93$) provided information on ecological interactions involving *H. madagascariensis*. Out of all defined interactions (Table 1), frugivory was the most reported ($N = 29$), followed by flower visitation ($N = 12$), and mycorrhizal association ($N = 8$). Competition, parasitism and myrmecochory were the least cited, with respectively only four, three and two studies mentioning those interactions. Neither amensalism nor interactions with endophytes have been recorded in this study, and 21 results included undefined interactions (5.3%). Finally, information on the potential of *H. madagascariensis* for forest restoration was obtained in 9.8% of the results ($N = 39$), including only one warning about its invasiveness and the threats it might pose to biodiversity.

### Distribution and habitat

Our results show that the natural distribution of *H. madagascariensis* spans across tropical Africa, Madagascar, and the Mascarenes (even though likely extinct on Réunion and absent from Rodrigues); and only introduced to Australia. It is found across tropical rainforests and savannahs and is native to 34 countries. Among the search results used to analyse its distribution and habitat, 57.6% ($N = 133$) recognized its pioneer status in the tropical forest succession, with the species typically dominating young secondary forests (0–12 years) (*Ndam & Healey, 2001*; *Hervé et al., 2015*) and declining afterwards

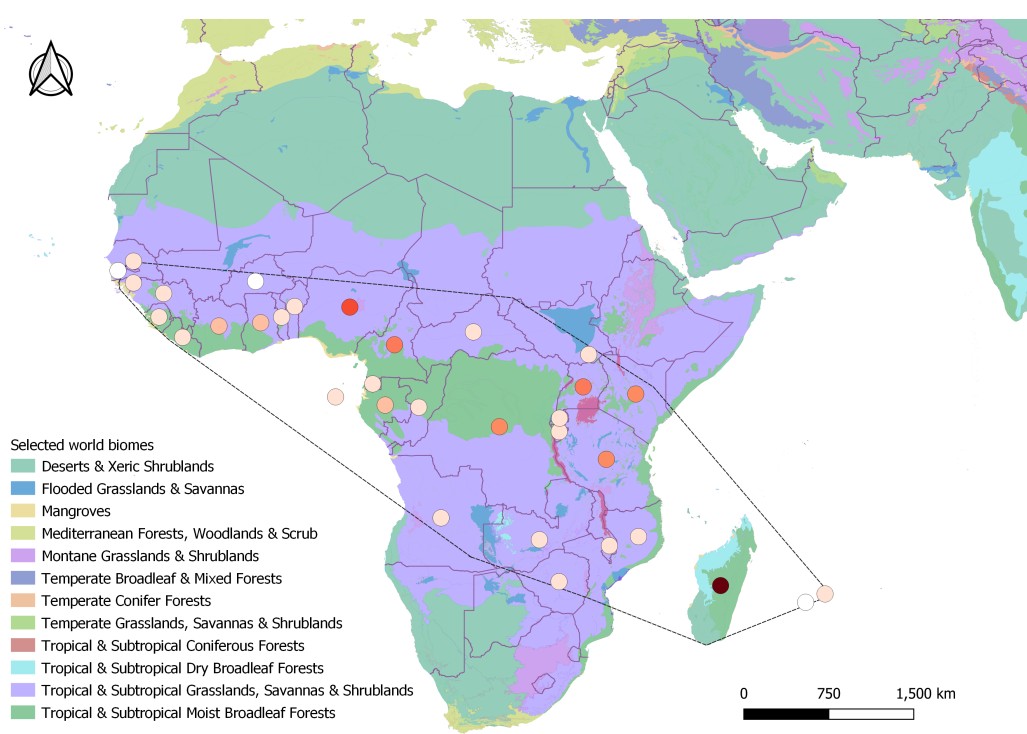

**Figure 2** Geographical distribution of studies retrieved under this review across *Harungana madagascariensis* native range (dashed line, $N = 398$). Studies spanned over 32 countries and two biomes. Circles represent study counts per country, colored as white (only GBIF record), beige (1–10 studies), orange (11–40), red (41–60) and brown (61–80 studies). Biomes were generated from Terrestrial Ecoregions of the World, from the World Wildlife Fund database (*Olson David et al., 2001*).

even though some individuals may survive up to 30–40 years (*Randriamalala, 2009*). In Australia, where it has been introduced, it is considered as an invasive species excluding native ones of unspecified ecology (*Humphries, Groves & Mitchell, 1991*), growing mostly after disturbances like in cyclone-damaged forests, as well as along forest fringes, roadsides and drains (*Vitelli & Van Haaren, 2001*). Even though it has also been said to be invasive in Mauritius (*Ragen, 2007*), all local Floras considered it native (*Bojer, 1837*; *Baker, 1877*; *Robson & Stevens, 1980*). *Harungana madagascariensis* together with *Vismia rufenscens* forms a clade related to the American *Vismia*, supporting its African origin (*Ruhfel et al., 2011*), and no studies reported the presence of ecotypes or genotypic variation within the species despite its very large geographical native range spanning over 12 million square kilometres (*BGCI et al, 2019*).

In all, nine broad ecosystems harbouring *H. madagascariensis* have been compiled from the search results and GBIF (2023) records. Forests and savannahs are the most cited ecosystems, with respectively 43.7% ($N = 475$) and 15.9%, ($N = 173$) of the citations compiled. Despite the fact that it grows primarily from coastal to montane forests, and from herbaceous to woody savannahs, it often also inhabits disturbed anthropogenic areas (including roadside, old quarry, mines, and nearby habitation) (14.7%, $N = 160$), wetlands (bordering creeks, lagoon, lakes, marshes, ponds, rivers and swamps) (11.2%, $N = 122$),

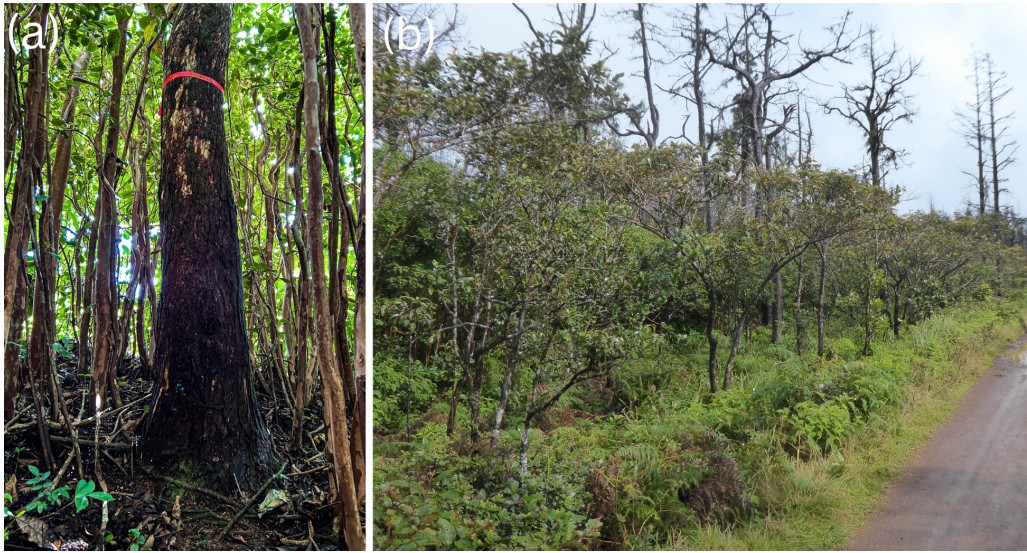

**Figure 3** **Habitats of *Harungana madagascariensis*.** (A) Tree trunk of *Harungana madagascariensis* within a stand of invasive alien species within its native range, showing its relatively large size (here about 30 cm in diameter) and fissured bark conducive to retaining water, relative to the alien invasive weeds around. (B) *Harungana madagascariensis* dominating regrowth following a major disturbance several years earlier. Photos by François Baguette.

fallows and cultivated land (9.1%, $N = 99$), agroforestry systems (2.1%, $N = 25$) and woodlands (1.8%, $N = 20$). As an heliophilous species, it is commonly found in forest edges and gaps (6.2%, $N = 67$) and constitutes a characteristic species of secondary and disturbed forests and savannahs (17.5%, $N = 190$) (Fig. 3).

With regards to habitat characteristics, 37.3% ($N = 148$) of our results included relevant information either on the elevation range, annual rainfall, temperature range, or the soil type where *H. madagascariensis* was observed, as well as 45 records from GBIF (2023). They show that *H. madagascariensis* grows in areas ranging from 10 to 2,467 m elevation and receiving an average of 740 to 12,500 mm of rain annually. Our results show however that *H. madagascariensis* grows mostly between 500 and 1,500 m elevation, and in areas receiving >1,500 mm of rain annually. It grows on a variety of soil types as defined by the *IUSS Working Group WRB (2022)* with a predominance on ferralsols, mostly yellow-red, red, red-yellow, and brownish tropical clay soils, that are well-drained and hydromorphic, but is also found in deep moderately drained soils (such as Haplic Alisols and Acrisols), on dystro-mollic Nitisols, ferric Luvisol, Arenosols and greyish alluvial soils restricted to alluvial plains and valley bottoms. As such, it grows mostly on poor, acidic and fragile substrate but can also be found on black deep humic soil and occasionally on soil of lateritic and volcanic origins.

## Ecological interactions

In all, we recorded 11 different interspecific ecological interactions between *H. madagascariensis* and 125 species (Table 1). Interactions recorded were grouped in
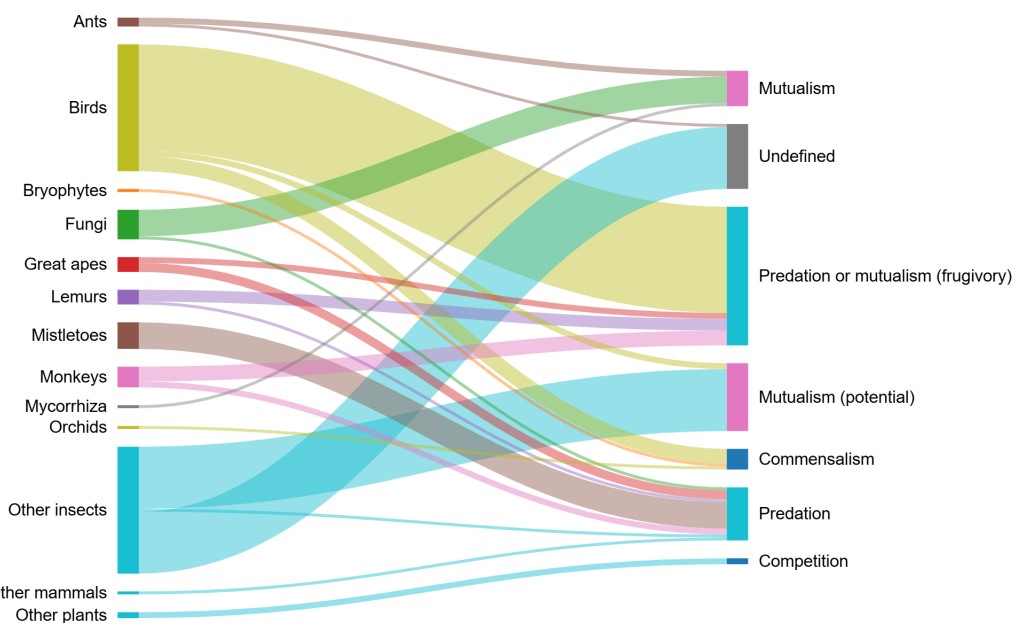

**Figure 4** **Interaction network showing the different groups of organisms interacting with *Harungana madagascariensis* (left), and the different types of interactions that they enter in (right).** Line thickness of the nodes represents the number of species interacting with *H. madagascariensis* and contributing to each interaction type. The list of all publications linked to each type of interaction is available in Table S2. Illustration made using SankeyMatic.com (https://sankeymatic.com/).

categories denoted as "+/+" for mutualistic interactions, "+/0" for commensalism, "+/−" for predatory interactions, "−/0" for amensalism, and "−/−" for competition. Other interactions which could not clearly be classified as above were recorded as "undefined". The main types of interactions recorded were "predation or mutualism" (frugivory), mutualism (potential) (flower visitation), predation (including florivory, folivory, parasitism, and stem damage), mutualism (including mycorrhizal association and myrmecochory), commensal interactions with birds, epiphytes, as well as saprophytic fungi and their host, and competition. *Harungana madagascariensis* was also used as a host for insects but without information on the nature of their interactions. The list of all species and their respective interactions with *H. madagascariensis* is provided in Table S4 and the network of interactions is illustrated in Fig. 4.

*Harungana madagascariensis* is known as a highly polliniferous and nectariferous species (*Akunne, Akpan & Ononye, 2016*). In all, 24 species from 12 genera and five families have been recorded visiting its flowers. Bees were by far the most diverse group of organisms recorded visiting flower, with 21 species from three families, (Apidae, Halictidae and Megachilidae). Two species of nectarivorous sunbirds, *Cinnyris venustus* and *C. reichenowi*, were also recorded as well as one species of mosquito, *Anopheles implexus*. No reptiles were recorded visiting flowers of *H. madagascariensis* despite being important plant pollinators and seed dispersers in the tropics and especially on islands. Out of those species, only

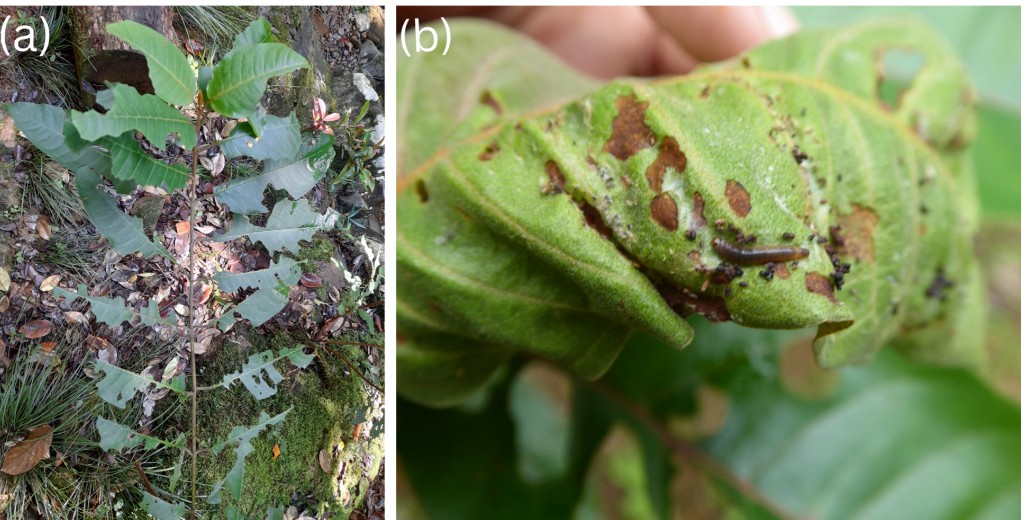

**Figure 5** **Herbivory on *Harungana madagascariensis*.** (A) Leaves of *Harungana madagascariensis* bearing marks by folivore. (B) Caterpillar feeding on a leaf of *H. madagascariensis*. Photos by François Baguette.

the sunbirds were identified as part of the pollinator network of *H. madagascariensis*. No studies looking at the role of insects as pollinators could be found.

Besides flower visitation, herbivory was another important interaction involving *H. madagascariensis* (Fig. 5) as 50 species were recorded throughout its range consuming its fruits mainly (frugivory), but also its leaves (folivory), flower buds (florivory), and stem. In all, 36 species of birds have been recorded feeding on *H. madagascariensis*, as well as 12 species of primates (five cercopithecid, three hominid, and four lemurs), one species of Elephantidae, and the Silk moth *Borocera marginepunctata* (Lasiocampidae). A total of 83% of those species consumed only fruits, while others consumed fruits and young leaves (3.7%), fruits, flower buds and young leaves (1.9%), only leaves (3.7%), or stem and leaves (1.9%). Only the Grauer's gorilla, *Gorilla beringei graueri*, was recorded feeding on *H. madagascariensis* but without specifying the part of the plant that was consumed. In addition, 22 species of insects have been recorded using *H. madagascariensis* as a host but without clear information on the nature of their interactions. Among those, Hemiptera, Hymenoptera, and Coleoptera were the most represented orders with respectively eight, five and five species recorded, followed by Phasmatodea (two), Lepidoptera (one) and Orthoptera (one).

Other types of interactions included commensal interactions with birds, epiphytes, and saprophytic fungi, mutualistic interactions with mycorrhizae and two species of ants (myrmecochory), as well as predatory ones with parasitic plant species and fungal diseases, and competition. Indeed, *H. madagascariensis* has been considered as a good phorophyte for orchids, ferns, fern allies and bryophytes throughout its range (*Malombe, 2009*; *Mangambu Mokoso de et al., 2015*; *Baider & Florens, 2022*), and even the species holding the highest diversity of mistletoes in a study conducted in the Bwindi Impenetrable Forest in Uganda

(*Kamusiime & Sheil, 2021*). Different types of fungi also grow on *H. madagascariensis*, some colonizing its roots and forming dense ecto-mycorrhizae, others saprophytic with some present exclusively on that species, and coelomycetes, such as *Idiocercus pirozynskii*, growing on its leaves (*Mulas & Rambelli, 1995*; *Wu & Sutton, 1995*; *Gibby, 2023*). On the other hand, *H. madagascariensis* has been found competing for light resources with heliophilous species such as *Psidium guajava* (an alien small tree in Africa) and *Baillonella toxisperma* (*Kouadio & Doucet, 2009*; *Otuoma et al., 2020*).

**Restoration potential**

In all, 39 publications contained information on the potential of *H. madagascariensis* for forest restoration. *Harungana madagascariensis* was recognized as a good restoration tool in the majority of them ($N = 38$, 97.4%), with a single publication considering it as a threat to biodiversity (2.6%). Within those the former 13 publications (33.3%) attributed its benefit to the heliophilous and pioneer trait of the species and its rapid growth rate (Fig. 6). Indeed, when colonizing a new area, *H. madagascariensis* can rapidly become dominant and form a closed canopy forest within four to six years (*Ndam & Healey, 2001*), growing at a rate 0.72 cm/year in trunk diameter and reaching 12 m high in five years (*Swaine & Hall, 1983*; *Manjaribe et al., 2013*). By doing so, it allows for a rapid shading of open areas, reducing the ability of potentially invasive species to grow. However, this rapid growth, assisted by the high network of mycorrhizae present on its roots and its capacity to reproduce vegetatively *via* root suckering (*Bellefontaine & Malagnoux, 2009*), also led some managers to control it, considering the species as invasive even in its native range (*Ragen, 2007*) and, in its introduced range, it has been reported to exclude native species establishment in recently opened areas (*Vitelli & Van Haaren, 2001*).

Other reasons recognized in the literature to use *H. madagascariensis* as a restoration tool included the following (in order of most to least common): nurse tree potential; capacity to improve or indicate soil fertility; capacity to compete with other heliophilous species; economic potential (*e.g.*, for honey production, timber, medicine and dye) ($N = 4$, 8.8% each); high seedling survival rate ($N = 3$, 6.7%); potential for low-cost restoration; capacity to suppress grasses; high resilience against disturbances (*e.g.*, fire and grazing); carbon sequestration potential ($N = 2$, 4.4% each); benefits for wildlife; phytoremediation potential; and supporting bio-control agent ($N = 1$, 2.6% each). Only one publication ($N = 1$, 2.6%) did not mention any specific reason. However, except for the survival rate of its seedlings, most of the other potential benefits of using *H. madagascariensis* for restoration either have not been quantified specifically (*e.g.*, nurse tree potential, ability to compete with heliophilous species), or have been estimated along with other species (*e.g.*, carbon sequestration, *Van Rooyen et al., 2023*).

## DISCUSSION

In this scoping review we identified 398 studies that provided information on *H. madagascariensis* habitat and distribution (as well as 4,379 records from GBIF), the ecological interactions it is involved in, and its potential roles for restoration. Our findings indicate a wide distribution across tropical Africa, Madagascar and the Mascarenes. We

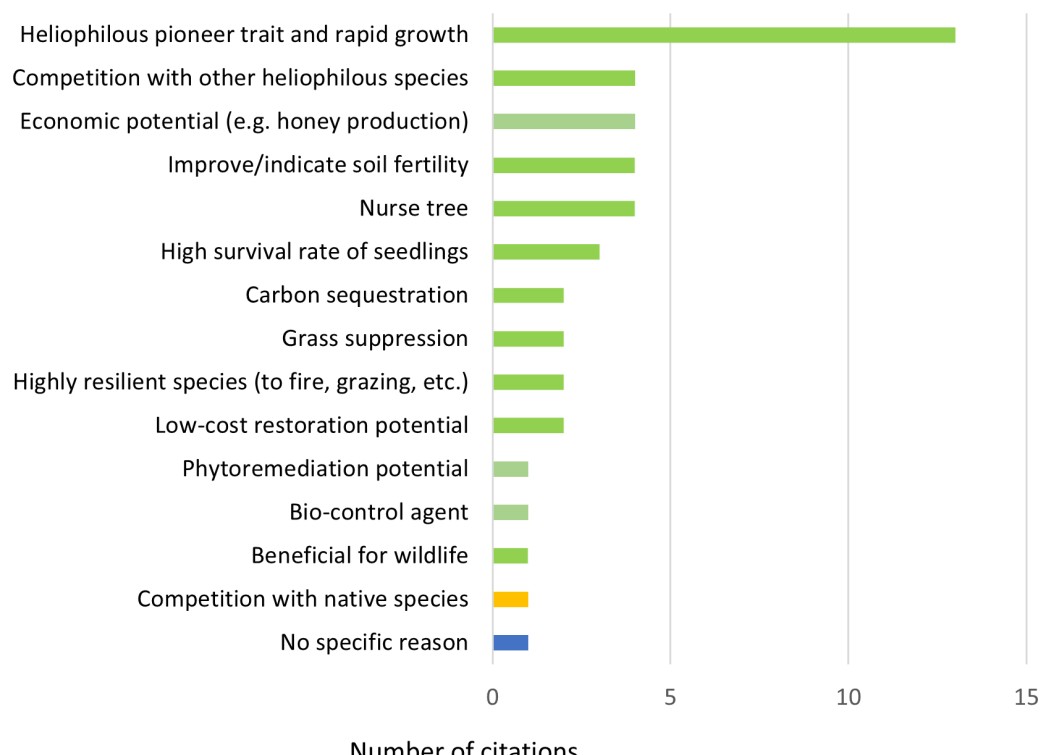

**Figure 6** **Reasons to support and oppose the use of *Harungana madagascariensis* in restoration projects.** Number of citations ($N = 45$) per reason given to support (in green) and oppose (in yellow) the use of *H. madagascariensis* in restoration projects. One publication (in blue) only considers the species as a restoration tool without mentioning any supporting reason. All reasons relate to the species in its native range. The list of all publications linked to each restoration potential is available in Table S2.

also found that *H. madagascariensis* interacts with at least 125 species over its range in different ways and is widely recognised as a good ecological restoration tool. However, against this background of positive influence, it has rarely been the focus of dedicated scientific research, and focussed studies are therefore needed to more fully characterise and precisely quantify its direct and indirect contributions to biodiversity conservation. To make our review of pioneer tree roles for biodiversity conservation and restoration more feasible and improve its potential usefulness, we decided to use *H. madagascariensis* as a model because of its wide native distribution but by doing so, our scoping review has some limitations. As such, our results might omit certain additional benefits or problems that other pioneer tree species (though generally less widespread than *H. madagascariensis*) could have or pose but which are absent or imperceptible in *H. madagascariensis*.

## Spatio-temporal distribution

Pioneer tree species are known for their ability to tolerate a wide range of environmental conditions (*Seidler & Bawa, 2001*), which allows them to grow in many different habitats and on a variety of substrates. In the case of *H. madagascariensis*, this review shows that its native range spans throughout tropical Africa, Madagascar and the Mascarenes, and that it

grows from coastal to montane forests, and from poor hydromorphic soils to deep humic soils. Indeed, due to its high association with mycorrhizae (*Rasoamampionona et al., 2008*; *Ramanankierana et al., 2013*) and its capacity to reproduce vegetatively *via* root-suckering (*Bellefontaine & Malagnoux, 2009*), *H. madagascariensis* can rapidly colonize disturbed ecosystems such as degraded forests (*e.g.*, exploited or damaged by natural weather events like cyclones), savannahs, and anthropogenic areas such as road sides, drains, and abandoned land. Due to the photosensitivity of its seeds (*Breyne, 1976*), it germinates and grows primarily in early successional and secondary forests, as well as in wetlands and savannahs, in other words in situations where light is abundant. Within late-successional forests, it typically grows in forest gaps and edges and despite the fact that it can dominate secondary forests up to ten years old, it is then progressively replaced by post-pioneer and late-successional species and therefore does not remain dominant in the longer term (past 10–20 years) (*Guelly et al., 2000*; *Foumier, Floret & Gnahoua, 2001*; *Hervé et al., 2015*), and it is not considered as one of the hyper dominant species of Africa (*Cooper et al., 2024*).

Even if some might argue that pioneer tree species generally compete with and sometimes even out-compete native forest species in its introduced range (*Vitelli & Van Haaren, 2001*) and even in its native range (*Ragen, 2007*; *Swinfield et al., 2016*), they can only do so during early forest succession stages or in specific non-forest ecosystems that suffered from disturbances and with species having similar resource requirements. Indeed, once they shade the ground, shade-tolerant species start germinating and the germination rate of heliophilous species like *H. madagascariensis* declines due to the reduced amounts of sunlight reaching the ground (*Swaine & Hall, 1983*; *Vroh & Yao, 2018*). Over the tropical forest succession dynamic, it is therefore normal that pioneer tree species dominate early-successional forest stages, to then gradually decline as more competitive species take their place. In other type of ecosystems however, and outside its native range, careful risk assessment should be implemented as their heliophilous trait and rapid growth rate could make them become invasive, competing with other light-demanding species of the native flora (*e.g.*, in Australia).

## Biological implications

*Harungana madagascariensis* plays a critical role in supporting biodiversity with at least 125 species interacting with it across its native range in one way or another, including endangered species such as chimpanzees and bonobos (*Yamakoshi, 1998*; *Trolliet et al., 2016*). The species particular importance lies in the fact that it brings its benefits over a large area (12,948,127.04 km$^2$ EEO) (*BGCI et al., 2019*) to which it is native and also over a wide range of ecosystem types where it naturally grows. The most cited ecological interaction benefiting biodiversity in which *H. madagascariensis* played a major role is herbivory. Indeed, virtually all parts of the tree constitute a resource for different organisms; its numerous small-fleshy fruits, rich in energy (*Atsalis, 1999*; *Donadeo, 2013*) being the most prevalent resource attracting many different species that in turn help disseminate its seeds and increase seed germination rate (*Bussmann & Lange, 2000*). However, no studies were found that quantified the relative frequency of herbivory on *H. madagascariensis* compared to other species nor measured whether it influences on seed dispersal of other species (for

example through attracting seed rain to the canopy shadow of *H. madagascariensis* trees) and to support herbivore communities.

The potential of *H. madagascariensis* to provide significant resources to pollinators (in the form of nectar and pollen) is also real as 24 species have been recorded visiting its flowers, but more studies are needed to capture the full extent of this ecological role because very few of the studies were specifically dedicated to flower visitation. Indeed, all species that were recorded visiting *H. madagascariensis*' flowers under this review have been extracted from studies treating of various other aspects (*e.g.*, studying pollen morphology of melliferous plants or the behavioural ecology of host selection of *Anopheles implexus*) but very few studies addressed specifically the question of pollination of this pioneer tree species or investigated flower visitor and pollen transport networks. Overall, *H. madagascariensis* was included in only two studies out of 12 that specifically addressed this topic, which identified sunbirds as part of flower visitation and pollen transport networks (*Nsor, 2014*; *Nsor, Godsoe & Chapman, 2019*). Out of eight studies identifying various species of bees visiting *H. madagascariensis*' flowers, only two specifically reported bees consuming nectar (*Latham & Mbuta, 2005*; *Akunne, Akpan & Ononye, 2016*) and one reported the presence of pollen of *H. madagascariensis* in honey (*Rasoloarijao, 2018*) but none of them studied their role in pollen transport networks.

Finally, this review also shows that other plant species benefit from the presence of *H. madagascariensis*, as it is known as a good phorophyte for epiphytic species such as orchids (*Baider & Florens, 2022*), ferns (*Mangambu Mokoso de et al., 2015*), as well as parasitic plants (*Kamusiime & Sheil, 2021*) and bryophytes (*Malombe, 2009*). Dedicated studies are however needed to more fully characterize epiphytic communities in relation to their phorophytes including *H. madagascariensis*, and overall, knowledge gaps on the extent to which pioneer trees foster epiphytes and parasitic plants still remain, especially in the context of forests facing high threats by invasive alien plants such as in Mauritius. Nevertheless, our review shows that pioneer tree species such as *H. madagascariensis* can provide a wide range of benefits on a large spatial scale in a very short period of time compared to other later successional species (*Piovesan & Biondi, 2021*; *Condit, 2022*).

## Management implications for biodiversity conservation

The increasing human pressures on forest ecosystems and biodiversity worldwide (*Chu & Yu, 2002*; *Cardinale et al., 2018*), make forest restoration increasingly important and urgent if we are to reach the 2030 UN Global Biodiversity Framework and its global targets on ecosystem restoration (*Aronson & Alexander, 2013*; *Gann et al., 2019*). Restoration efforts must therefore be substantially increased and for that, effective solutions to reduce forest restoration costs are much needed (*Crouzeilles et al., 2020*; *Cook-Patton et al., 2020*). Current restoration projects often use radically opposed methodologies, which at the extremes, include some aiming to maximise plant diversity and are resource intensive through high maintenance that require capacity and funding (up to 9,922$/ha), while others promote methods that are less intensive and cheaper such as the nucleation and assisted natural regeneration methods (between 638 and 4,654$/ha) (*Elliott, Blakesley & Hardwick, 2013*; *Campanhã Bechara et al., 2021*). Even though this variety of methodologies

can help address different local issues, science-based and cost-effective projects still need to be deployed more widely to effectively conserve biodiversity and restore much larger areas than is currently the case (*Perring et al., 2015*; *Mansourian, Dudley & Vallauri, 2017*; *Lewis, 2022*; *Marshall et al., 2022*).

One way to move towards more science based and cost-effective restoration projects is to include common tree species, including species like *H. madagascariensis* in restoration projects. Indeed, despite the fact that common tree species constitute only 2.2% of tropical forest tree species, they account for 50% of all individual trees that structure those forests (*Cooper et al., 2024*). As such, they fit the 'rare is common, common is rare' natural pattern (*McGill, 2010*) also documented in many other taxa (*McGill et al., 2007*; *McGill, 2010*; *Henderson & Magurran, 2010*; *Baldridge et al., 2016*; *Enquist et al., 2019*) and constitute a critical component of natural tropical forests. Including such species further in forest restoration would therefore align ecosystem-level biodiversity and function closer to natural and more stable situations while substantially reducing restoration costs hence increasing our chances to upscale forest restoration worldwide. Although this does not detract from the importance of rare species, it puts emphasis on the importance to incorporate pioneer species and their ecological roles in ecological restoration if we are to optimize our response to future environmental change and support the efforts to stop biodiversity loss and enhance ecosystem restoration and function worldwide.

In line with the above, our study underscores the disproportionately large role that pioneer tree species can have towards biodiversity conservation and ecological restoration, despite the fact that they live for a much shorter time period compared to later successional species. This situation is particularly true for *H. madagascariensis* because Africa has a much lower tree diversity compared to tropical America and Asia, including of pioneer tree species (*Swaine & Hall, 1983*; *Pearson et al., 2002*; *Parmentier et al., 2007*; *Goodale et al., 2012*; *Sosef et al., 2017*). Major advantages to use the tree as a tool for ecological restoration projects have been acknowledged, such as its rapid growth, rapid maturation, and its attractiveness to a wide range of frugivores that can enhance seed dispersal (*Swaine & Hall, 1983*; *Bellefontaine & Malagnoux, 2009*; *Konersmann et al., 2022*). It shows that pioneer trees like *H. madagascariensis* therefore constitute a real asset to lower restoration costs and upscale projects globally, while also benefiting numerous other species that naturally interact with it. However, despite all those benefits, the species is still somehow considered as invasive by some conservation practitioners based on old reports that either consider it as such without reference (*Ragen, 2007*) or recommend its control based only on the observation that "it grows like an introduced species" (*Strahm, 1993*). Examples of such control come from Mauritius where trees are cut and uprooted (*Florens & Baider, 2013*), and more recently lopped or ring-barked (FMMP Baguette, C Baider, FB Vincent Florens, pers. obs., 2022–2024).

## Implications for reforestation and economic development

Reforestation in the tropics is considered as another important restoration tool for climate change mitigation (*Canadell & Raupach, 2008*) and biodiversity conservation (*Harrison, Wardell-Johnson & McAlpine, 2003*; *Kemppinen et al., 2020*; *Andres et al.,*

*2023*). However, reforestation at a landscape scale is complex and challenging (*Lamb, 2014*), especially given its cost (*Silva & Nunes, 2017*; *Benini & Adeodato, 2017*) and the increasing competition for land (*Alves-Pinto et al., 2017*). Typically, reforestation projects using exotic species have been highly favored for a variety of economic, logistic and environmental reasons (*Davidson, 1989*; *Mehari, 1996*; *Richardson, 1998*; *FAO, 2010*), providing both socio-economic and ecological benefits (*Senbeta & Teketay, 2001*; *Le et al., 2012*; *Randriambanona, Randriamalala & Carrière, 2019*; *Brancalion et al., 2020*; *Boissière et al., 2021*; *McElwee & Nghi, 2021*). However, negative impacts of this practice are also increasingly being recognized (*Dodet & Collet, 2012*; *Wang, Wang & Wang, 2013*), and therefore more cost-efficient and ecologically appropriate methods are now emerging (*Hanson et al., 2015*). The use of native species for reforestation project has been recognised as a solution (*Butterfield, 1995*; *Stimm et al., 2008*; *Lampela et al., 2017*; *Jensen et al., 2021*) but many challenges still need to be addressed to implement it at large scales (*Broadhurst, Waters & Coates, 2017*; *Nunes et al., 2020*) as the practical sylviculture of many native species remains poorly known (*Fine, 2002*; *Onyekwelu, Stimm & Evans, 2011*; *Le et al., 2012*).

Based on this context and on our results, *H. madagascariensis* could serve as a potential candidate for native reforestation projects in tropical Africa. Indeed, its wide distribution, high survival and germination rates, and pioneer status (*Bussmann & Lange, 2000*; *De Gouvenain, Kobe & Silander, 2007*; *Manjaribe et al., 2013*) are promising traits for reforestation projects. It is also known to be fire-tolerant (*Orwa et al., 2010*; *Hill, 2018*) and resistant to heavy grazing and trampling (*Hurault, 1998*), but more studies are needed to test these biological characteristics (*Olupot, 2004*; *Hill, 2018*) relative to other (though less widespread) pioneer tree species that grow in tropical Africa (*Ashton & Hall, 2011*; *SilanderJr, Bond & Ratsirarson, 2024*) for large-scale reforestation (*Negash, 2021*), and identify potential genetic variation or ecotypes in the species as no information on this aspect has been found in the literature (*Schoch et al., 2020*). Finally, caution should be used to avoid planting the species where it would not naturally occur, so as to limit possible risks of damaging critical habitats (*Culbertson et al., 2022*) and avoid risking losing critical biodiversity and ecosystem functionality (*Bond, 2016*; *Parr, Beest & Stevens, 2024*).

Additionally, another important aspect that would deserve further study is the social acceptation of *H. madagascariensis* for reforestation as community engagement is critical for yielding success (*Baynes et al., 2015*; *Herbohn et al., 2023*; *Bayne & Grant, 2024*). In this review, 262 papers have been identified treating on the medicinal properties of *H. madagascariensis,* including against various infectious diseases as well as skin and heart problems (*Happi et al., 2020*). Twenty-nine other studies also highlighted its economic importance, be it for the food (*e.g.*, honey production) and dye industries, the production of firewood and charcoal, the production of poles and planks, or as fuel in local metallurgy (*Lewis, 1986*; *Schure et al., 2009*). However, none of those benefits for communities were evaluated in the context of reforestation and therefore this knowledge gap remains to be filled. Nevertheless, those additional benefits enhances the species value in addition to its potential to promote faster, and cheaper hence more sustainable and wider-scale ecological

restoration that is also directly beneficial, through the provision of various resources, to many co-occurring species, including threatened ones.

## CONCLUSION

Using *H. madagascariensis* as a model, we showed that pioneer tree species significantly contribute to support biodiversity, providing resources to support many species over an extended area that can reach millions of km$^2$, and hosting important epiphyte communities, as well as parasitic plants, insects, and fungi species. As they do so in a very short time period after a disturbance compared to later-successional trees, and due to their high resilience to disturbances, rapid growth, and ability to compete with other heliophilous species (which are typically invasive alien species) among other benefits, they constitute a key species to foster better ecological restoration at lower costs. However, pioneer tree species have rarely been the focus of scientific research and dedicated studies are therefore needed to more fully quantify their direct and indirect contributions and benefits to biodiversity conservation. The extent to which they reduce ecological restoration costs would however depend much on which restoration approach their use in restoration projects would be compared with. Where non-pioneer species are favoured in restoration projects over pioneers, we hope that this study will provide the necessary incentives to conservation managers and restoration project funders to henceforth pay better attention to the importance of restoring ecological function and more natural and sustainable systems than is currently the case. The use of evidence-based management should promote the use of species like *H. madagascariensis* in restoration projects at a larger scale, allowing for more efficient, more successful and sustainable and more cost-effective conservation projects.

## ACKNOWLEDGEMENTS

The authors are grateful to the partners of this project, especially the University of Mauritius for its administrative support, Mr. Owen L. Griffiths and Mrs. Mary-Ann Griffiths for their permission to access Mt. Camizard and CIEL Group for their permission to access the Ferney Conservation Area. We would also like to thank the editor, Viktor Brygadyrenko, and the anonymous reviewers for their comments that improved the manuscript.

### Funding

This project was supported by the Agence Française de Développement (AFD) under the VARUNA Biodiversité programme (Project 22-SB3004) managed by Expertise France. The funders had no role in study design, data collection and analysis, decision to publish, or preparation of the manuscript.

### Grant Disclosures

The following grant information was disclosed by the authors:
The Agence Française de Développement through the Varuna Biodiversité programme (Project 22-SB3004).

## Competing Interests

The authors declare there are no competing interests.

## Author Contributions

- François M.M.P. Baguette conceived and designed the experiments, performed the experiments, analyzed the data, prepared figures and/or tables, authored or reviewed drafts of the article, and approved the final draft.
- Cláudia Baider conceived and designed the experiments, analyzed the data, prepared figures and/or tables, authored or reviewed drafts of the article, and approved the final draft.
- F.B. Vincent Florens conceived and designed the experiments, prepared figures and/or tables, authored or reviewed drafts of the article, and approved the final draft.

## Data Availability

The raw data are available in the Supplementary Tables.

## Supplemental Information

Supplemental information for this article can be found online at http://dx.doi.org/10.7717/peerj.19458#supplemental-information.

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
