# Peer review of "A scoping review of tropical pioneer trees’ roles for restoration and conservation management: Harungana madagascariensis (Hypericaceae) a widespread African species as a model"

_PeerJ, doi:10.7717/peerj.19458_

## Round 0.1 · original submission · Major Revisions

Dear authors, I ask you to listen to the very useful comments of the reviewers. I also recommend making the article more informative. The figures are of very low quality, some of them (Figure 7) are uninformative. I believe that you will be able to improve the manuscript and it will be approved by the reviewers.

Reviewer 1 ·

Basic reporting

The authors have submitted a manuscript, scoping in nature, that reviews what is known of the basic biology and ecology of a pioneer tree species, Harungana madagascariensis, which has a broad distribution across tropical Africa and the nearby Indian Ocean islands. The manuscript broadly surveys the literature and examines GBIF records to characterized the distribution of the species across its native range. This contrasts with typical manuscripts reporting research on experiments, data analyses, and/or modeling. I find that the manuscript overall is generally well-written and clear.

Experimental design

A. General comments:

!. Clearly, a primary motivation for this a manuscript is that most of the reforestation or afforestation efforts across the tropics rely on exotic species with well-known sylvicultural practices, like Eucalyptus spp. and tropical/subtropical pines, while the practical sylviculture or even the basic biology and ecology of native species remain poorly known (e.g. Onyekwelu et al. 2011, Le et al. 2012, Fine 2002, and many others). By relying on exotic species, one risks reducing biodiversity overall while converting native tropical ecosystems into alien ecosystems. The authors provide an important summary of what is known about the basic biology and ecology of Harungana madascariensis, and they should be congratulated. But there are some critical components missing here too.

2. There is no mention of practical silviculture, which is critical if one is to be serious about using this or any other pioneer species for reforestation or afforestation. There is some general wording in the section on Management Implications, but no specifics on how this would apply to the species and systems in question here. Silviculture is a complex, species and system specific problem not easily addressed by generalities, and includes everything from seed sourcing, developing nurseries to raise seedling and saplings, conducting provenance feasibility trials, establishing and managing plantings in diverse environments, and engaging locals (via social forestry or otherwise) across the impoverished rural communities that span the native range of Harungana madagascariensis, and there is the cost and the necessary infrastructure associated with all of this. Is the goal simply to plant forests of Harungana without addressing other social needs of the rural poor, e.g. timber, firewood, charcoal, etc. In contrast, this is where everything is in place across the tropics for Eucalypts and pines: the silvicultural knowledge and infrastructure is in place, costs and management is well known and social value to rural peoples are addressed: i.e. sources for timber, firewood, charcoal, etc. Maybe there is some hope at a small scale with studies like that of Manjaribe et al. 2013. But, the authors should at least spend a bit of time on a reality check of the challenges involved, given the prime objective of the manuscript “forest restoration and identify eventual knowledge gaps.” See also Lamb 2014 for a general discussion of the global challenges of reforestation, and Culbertson et al. 2022 for a detailed discussion of the many challenges on the ground of reforestation, specifically for Madagascar, along with the critical sylvicultural knowledge gaps that require further study. One of the questions for Madagascar in particular, is why Harungana madagascariensis, as a wide-ranging native pioneer species, has shown so little tendency to spread anywhere across the vast open grassy systems that dominate much of Madagascar today? This is in contrast with, for example, the native Ravenala madagascariensis which easily spreads as a pioneer across the wetter ecosystems of Madagascar (cf. Geldenhuys 2013). The challenges of using untested native species for large-scale reforestation do merit some acknowledgement and discussion.

3. The authors probably might also acknowledge that Ecalyptus- or Pinus-based reforestation is not necessarily all bad. Several studies have shown that one can find native woody species and associated biodiversity regenerating under Eucalyptus and Pinus canopies (depending on the planted species selection and canopy structure); see e.g. Randriambanona et al. 2019, Senbeta and Teketay 2001.

4. One of the questions the reader may have is: why choose Harungana madagascariensis? The authors address this in the Abstract Conclusions: it is “one of the few native pioneer tree species of tropical Africa.” But a quick check shows that this is a misleading statement. For example, see table 12.1a in Ashton and Hall 2011, which lists some 20 pioneer (or shade intolerant) tree species for West and Central Africa, or Silander et al. 2024 Table 1, which lists some 40 pioneer or open ecosystem woody species for Madagascar alone. Now Harungana may offer advantages as candidate pioneer species for reforestation across Africa because of its broad distribution, and now that the authors have summarized what all is known about the biology and ecology of the species provides perhaps a leg up. But clearly the authors need to acknowledge that that are indeed many other pioneer or open ecosystem candidate tree species native to tropical Africa.

5. The authors should reduce some redundancy or unnecessary wording and figures in the manuscript. For example, lines 133-137 list all of the synonyms or alternative spelling for Harungana madagascariensis, when one could simply write “Harungana madagascariensis and all of its synonyms and alternative spellings” (and if needed, put this list in an online appendix). Much of the Discussion section (i.e. lines 383-576) repeats much of what was already stated in the Results section. Also cut figures 6 and 8 as they really don’t add anything critical to the manuscript and its message.

6. There are several places where the authors state that a number of studies have shown x, y, z but then just list the numbers of studies without clearly linking these to specific references. For example, see lines 352-356: “42 publications contained information on the potential … for forest restoration….13 publications…attributed its benefit to …pioneer trait … rapid growth rate…” The same is true for figures 4 and 7 and Table 1. I assume that some or all of these studies referenced come from the long list of 398 papers in table S2, rather than from the References section of the manuscript alone. It seems that as a minimum these referenced publications should be linked to an online ‘S’ table that lists (with subsections as needed and parallel to Table 1) the referenced studies that point to each cited processes or patterns mentioned in the text of the manuscript.

Validity of the findings

B. Specific comments:
1. The authors have been quite exhaustive in finding literature on the biology and ecology of Harungana madagascariensis, but they surprisingly missed at least one key reference: de Gouvenain et al. 2007, which discusses in some detail seedling growth, survival and performance of Harungana madagascariensis, in comparison with 3 other native tree species, across a gradient of light (i.e. reflecting competition for light) and soil moisture conditions. It does make me wonder what else might have been missed or deleted.

2. The authors make little mention of fire tolerance, grazing/browsing tolerance, or competition with grass and other species encountered during early succession, or under conditions of establishing seedlings or saplings in a reforestation initiative. They simply say on lines 372-373: “…potential for low-cost restoration; capacity to suppress grasses; high resilience against disturbances (e.g. fire and grazing)” without references. This is the only place in the manuscript where fire or grazing are mentioned. Clearly there are critical gaps in knowledge on the performance of Harungana madgascariensis under different fire regimes (especially in fire-prone landscapes), the impacts of large grazers and browsers (especially in African savannas or with domestic livestock in rural villages), and competitive abilities under a range of different site conditions. This gap in knowledge certainly merits mention in the manuscript.

3. Figure 2, Geographical Distribution (at least in my copy of the ms.), is of rather poor quality and resolution and not particularly useful. I would suggest that rather than a low-resolution map of the world, that the authors use a map of Africa and the Indian Ocean islands (with maybe an inset for the Mascarenes) that encompasses the native range of Harungana madagascariensis with dots showing the location of all GBIF data points. One could take this further and show a convex hull of the points yielding an approximate native distribution of the species. One could even take this a bit further with a projected distribution of the species using widely available software like MaxEnt, which would be easy, given all of the covariate data on distributions that the authors have assembled. As for locations of the species outside the range, one could just list these separately, rather than showing this on a global map. If one wants to include biomes on the map (which is entirely too busy as presented) one could just have the relevant biomes here, but this may still be too busy. But then, how many biomes to map, as there is little agreement as to not only where the boundaries are (which are inherently fuzzy) but also what to include, e.g.: just forest vs. savanna; or rain forests, moist forests, monsoonal forests, dry forests, woodlands, wooded savannas, grass savannas, thickets, etc. Or maybe the “9 broad ecosystems” in which the species is found?

4. Another issue to mention briefly is that, given the broad distribution of Harungana madagascariensis across Africa and Indian Ocean islands, and across broad elevational and rainfall gradients, what is known, if anything, about ecotypic or genotypic variation in the species? This is one of the motivations for doing provenance trials of seedling or sapling materials before planting these out in any large-scale reforestation projects.

5. Some mention of social forestry is merited. It is fine to list the potential for medicinal uses, honey cultivation, firewood, charcoal and wood, etc. but what do the local think of Harungana .
madagascariensis versus other tree species? For reforestation project across rural Africa engaging locals is a critical component for yielding any degree of success (e.g. Baynes et al. 2015).

6. One final note of caution worth mentioning, given the push in recent years to reforest the tropics: If Harungana madagascariensis or indeed any pioneer (or exotic) species is to be used in broad afforestation projects: caution should be used in planting forests in savanna biomes, or one risks loss of critical biodiversity and ecosystem functionality in extant savanna systems (see: Bond 2016, Parr et al. 2024). Another example is the grassy ecosystems in the Mascarenes that are considered critically threatened habitats (along with the species therein) that were maintained by the now extinct giant tortoises (cf. Griffiths et al. 2010).

References:
Ashton, M.S. and Hall, J.S., 2011. Review the ecology, silviculture, and use of tropical wet forests with special emphasis on timber rich types. Silviculture in the Tropics, pp.145-192.
Baynes, J., Herbohn, J., Smith, C., Fisher, R. and Bray, D., 2015. Key factors which influence the success of community forestry in developing countries. Global Environmental Change, 35, pp.226-238.
Bond, W.J., 2016. Ancient grasslands at risk. Science, 351(6269), pp.120-122.
Culbertson, K.A., Treuer, T.L., Mondragon‐Botero, A., Ramiadantsoa, T. and Reid, J.L., 2022. The eco‐evolutionary history of Madagascar presents unique challenges to tropical forest restoration. Biotropica, 54(4), pp.1081-1102.
de Gouvenain, R. C., Kobe, R. K., & Silander, J. A. 2007. Partitioning of understorey light and dry-season soil moisture gradients among seedlings of four rain-forest tree species in Madagascar. 23(5), pp. 569–579.
Fine, P.V., 2002. The invasibility of tropical forests by exotic plants. Journal of Tropical Ecology, 18(5), pp.687-705.
Griffiths, C.J., Jones, C.G., Hansen, D.M., Puttoo, M., Tatayah, R.V., Müller, C.B. and Harris, S., 2010. The use of extant non‐indigenous tortoises as a restoration tool to replace extinct ecosystem engineers. Restoration Ecology, 18(1), pp.1-7.
Geldenhuys, C.J., 2013. Recovery of forest biodiversity by natural ecological processes through native or alien tree stands.
Lamb, D., 2014. Large-scale forest restoration. Routledge.
Le, H.D., Smith, C., Herbohn, J. and Harrison, S., 2012. More than just trees: Assessing reforestation success in tropical developing countries. Journal of Rural Studies, 28(1), pp.5-19.
Onyekwelu, J.C., Stimm, B. and Evans, J., 2011. Review plantation forestry. Silviculture in the Tropics, pp.399-454. (and the book this chapter comes from)
Parr, C.L., Te Beest, M. and Stevens, N., 2024. Conflation of reforestation with restoration is widespread. Science, 383(6684), pp.698-701.
Randriambanona, H., Randriamalala, J.R. and Carrière, S.M., 2019. Native forest regeneration and vegetation dynamics in non-native Pinus patula tree plantations in Madagascar. Forest Ecology and Management, 446, pp.20-28.
Senbeta, F. and Teketay, D., 2001. Regeneration of indigenous woody species under the canopies of tree plantations in Central Ethiopia. Tropical Ecology, 42(2), pp.175-185.
Silander Jr, J.A., Bond, W.J. and Ratsirarson, J., 2024. The grassy ecosystems of Madagascar in context: Ecology, evolution, and conservation. Plants, People, Planet, 6(1), pp.94-115.

Reviewer 2 ·

Basic reporting

None.

Experimental design

L168-175: For data collection, did you consider conference papers, project reports? Justify the inclusion of papers that do not have rigorous peer review.

Additionally, explain what are others?

Validity of the findings

I suggest combining the following four figures (Figure 3, Figure 5, Figure 6 and Figure 8) for a better organization of the manuscript, so as not to saturate the number of figures.

Additional comments

The study presents a review about the potential of pioneer tree species (Harungana madagascariensis) for biodiversity conservation and forest restoration. In my opinion, the study is well developed, and the results meet the stated objectives supported by a solid methodology.

---

## Round 0.2 · accepted · Accept

Dear Dr. Baguette, I am pleased to inform you that your article has been accepted for publication. I hope that you will continue to prepare such high-quality and interesting articles on this topic.

Reviewer 1 ·

Basic reporting

The authors have done an excellent job revising the manuscript and have addressed all of the comments that I made in my review of the prior version of the manuscript. The net result is a much better manuscript and the authors should be congratulated on a job well done. In consequence I have no further comments to make about the manuscript.

Experimental design

Fine.

Validity of the findings

Excellent

Additional comments

None

Reviewer 2 ·

Basic reporting

The study presents a review about the potential of pioneer tree species (Harungana madagascariensis) for biodiversity conservation and forest restoration. In my opinion, the study is well developed, and the results meet the stated objectives supported by a solid methodology.

Experimental design

None

Validity of the findings

None

Additional comments

None